# Higher-Order Belief Propagation Correction Decoder for Polar Codes

**DOI:** 10.3390/e24040534

**Published:** 2022-04-11

**Authors:** Meng Zhang, Zhuo Li, Lijuan Xing, Xin Liao

**Affiliations:** The State Key Laboratory of Integrated Services Networks, Xidian University, Xi’an 710071, China; mzhang_218@stu.xidian.edu.cn (M.Z.); ljxing@mail.xidian.edu.cn (L.X.); lxin2526@126.com (X.L.)

**Keywords:** polar code, belief propagation decoding, post-processing, higher-order correction

## Abstract

Belief propagation (BP) decoding for polar codes has been extensively studied because of its inherent parallelism. However, its performance remains inferior to that of successive cancellation list decoding (SCL) due to the structure of the decoding graph. To improve the block error rate (BLER) performance, the BP correction (BPC) decoding, a post-processing scheme that corrects prior knowledge of the identified code bit, improves convergence by executing additional iterations on the failed BP decoder. Moreover, the BPC decoder demonstrates a better decoding performance than the BP-based bit-flipping decoder. Nevertheless, the additional decoding attempts lead to increased latency. In this article, a modified BPC decoder is proposed to reduce the number of decoding attempts by redefining the correction rules. A new metric is designed to effectively identify the corrected location. Numerical results show that the proposed modified BPC decoder achieves a slight improvement in BLER compared with the original BPC, with a dramatic reduction in average complexity. Furthermore, a higher-order version, named MBPC-Ω, is extended to further improve the performance, where the Ω is the maximum correction order. Numerical results show that the higher-order modified BPC achieves a similar BLER performance to existing multiple bit-flipping BP decoders but has around half the latency overhead. In addition, the proposed MBPC-2 decoder performs better than the cyclic redundancy check-aided SCL (CA-SCL) decoder with list size 4 and is slightly worse than the CA-SCL with list size 8 in high signal-to-noise ratio (SNR) regions but with significant decoding latency reduction.

## 1. Introduction

Polar code [1] is a mathematically proven capacity-achieving coding scheme and has specific construction methods and linear encoding and decoding complexity, which have been selected as the coding scheme in fifth-generation enhanced mobile broadband (eMBB) control channels [2]. A successive cancellation list (SCL) decoder with a cyclic redundancy check (CRC) (CA-SCL) [3,4] obtains comparable error performance with that of the state-of-the-art LDPC code. However, it suffers from high latency due to its intrinsic serial structure. Conversely, the belief propagation (BP) decoder [5,6] performs the decoding process in a parallel manner, therefore garnering more and more attention as a low-latency implementation method.

However, the original BP decoder has a worse block error rate (BLER) performance than that of the SCL decoder. Therefore, several BP-based decoding algorithms have been proposed to enhance BLER performance. By considering different representations obtained using graph permutations, the multi-trellis scheme, first proposed in [7], was further improved in [8,9,10] with different permuted patterns, resulting in better BLER performance, but its throughput subsequently dropped. In [11], a BP list (BPL) decoder performed *L* independent BP decoders on different permuted versions of the factor graph to yield a valid estimate. Additionally, the selection strategy of *L* permuted factor graphs which have the least number of girths was designed in [12], resulting in better performance and lower average decoding latency. Post-processing is another effective strategy to improve the BLER performance. A random perturbation-based post-processing method was developed in [13] to address different types of BP errors and to improve convergence. The noise-aided BPL decoder, combined with random noise injection and list decoding, introduced in [14], has a much better performance than the SCL and is a bit behind the CA-SCL decoder in terms of BLER performance. Inspired by the SC bit-flipping decoder, the BP flipping (BPF) decoder, as a post-processing method, was first proposed in [15] and was then improved in [16,17,18]. The BPF decoder, performing bit-flipping on error-prone information bits, achieved a comparable BLER performance with that of the SCL decoder with a small list size. The flipping set which is constructed by a convolutional neural network-based imitation learning scheme was presented in [19], such that useless flipping attempts are avoided. [20] proposed a noised-aided BPL bit-flipping decoder which is superior to the CA-SCL decoder at the cost of numbers of decoding iterations. In [21], a low-complexity two-level post-processing algorithm was proposed to modify the false converged error efficiently in the BPL decoder. In [22], the correction performed on code bits was studied. When the BP decoder failed, the correction operation performed as an external perturbation used to improve the convergence, named the BPC decoder. By correcting the unreliable code bits, i.e., assigning ±∞ to their log-likelihood rate (LLR) and performing additional iterations, the decoder is expected to converge correctly. For identical additional decoding attempts, the BPC decoder shows better BLER performance than the BPF decoder. However, without benefiting from prior knowledge about the identified code bit, assignments on both sides lead to excessive decoding attempts, resulting in relatively high decoding latency overhead.

In this article, the BPC decoder is modified by redefining the correction set based on a new metric and the correction operation based on the predefined threshold, abbreviated as the MBPC decoder. Then, the proposed MBPC decoding is extended to a higher-order scheme, named MBPC-Ω, where Ω code bits are corrected in one decoding attempt. The simulation results show that the MBPC decoder shows a slight performance gain but has less complexity and latency than the original BPC decoder. The higher-order MBPC decoders are verified to considerably improve the BLER performance. For polar codes of length N=2048 with 24-bit CRC termination, when $\mathrm{BLER}=10^{-3}$, the MBPC-2 decoder achieves around a 0.9 dB performance gain compared with the original BP decoder. Compared with existing BP-based multiple flipping decoders, the MBPC-2 performs a similar BLER but approximately half iterations. At $\mathrm{BLER}=10^{-5}$, the MBPC-2 decoder achieves a gain of 0.2 dB compared with the CA-SCL decoder with L=4, at a latency overhead saving of 95%. The contributions of this article are summarized as follows.
Different from using the stopping tree to reduce the searching range in the original BPC decoder, a new metric is proposed, which combines the reliability of code bits and the size of the stopping tree to which it belongs, to construct the correction set effectively.We refine the correction operation in the original BPC decoder. Setting a threshold V, an identified code bit is only corrected to its opposite side when its reliability is lower than V. The decoding latency is reduced by avoiding invalid decoding attempts, especially in low signal-to-noise ratio (SNR) regions. In addition, the refresh operation is modified by re-initializing the massage to the initial state of the original BP rather than the end state to remove the dispensable store procedures.The MBPC-Ω decoder is proposed to further improve the BLER performance, where the decoder is executed on correction sets of increasing order. The correction set $S_{\Omega }$, in which each element contains Ω code bits, is established nested based on the preorder set $S_{\Omega -1}$ and the corresponding decoding result.
The remainder of this article is organized as follows. Section 2 provides brief statements of the polar code, the BP decoder, and the BPC decoding algorithm. Section 3 analyzes the metric and provides the proposed MBPC decoding algorithm. The higher-order MBPC decoder is presented. The simulation results are presented in Section 4. Finally, Section 5 concludes the article.

## 2. Preliminaries

### 2.1. Polar Codes

Polar codes are linear block codes based on the channel polarization phenomenon. After channel combination and splitting, *N* channel copies are converted to *N* polarized sub-channels, where the capacity of each sub-channel approaches 1 or 0 as *N* increases. The polar codes select *K* reliable sub-channels to transmit information bits, and the remaining sub-channels transmit frozen bits. The sets of information indices and frozen indices are denoted by I and its complement $I^{C}$, respectively. For shorthand, we use $a_{i}^{j}$ to denote the vector $\left(a_{i},a_{i+1},...,a_{j}\right)$. The codeword $x_{1}^{N}$ can be mapped by the generator matrix $G_{N}$:(1)$$x_{1}^{N}=u_{1}^{N}G_{N},$$
where $N=2^{n}$, $G_{N}=R_{N}F^{⊗n}$. The $F^{⊗n}$ is the *n*-th Kronecker power of $F=\left(\begin{array}{cc} 1 & 0\\ 1 & 1 \end{array}\right)$ and $R_{N}$ is a N×N bit-reversal permutation matrix. The source vector $u_{1}^{N}\in {\left\{0,1\right\}}^{N}$ includes both information and frozen blocks.

### 2.2. Belief Propagation Decoder

BP decoding for polar codes is an iterative decoding algorithm based on the (n+1)-stage factor-graph realization of $G_{N}$. Two types of LLR messages are involved: right-propagating $R_{i,j}$ and left-propagating $L_{i,j}$, where *i* denotes the stage and *j* denotes the row index. The four nodes connecting two stages constitute the processing elements (PEs). The architecture of a PE is shown in Figure 1. Thus, the iteration rule can be represented by Equation (Equation 2):
(2)$$\left\{\begin{array}{c} L_{i,j}=f\left(L_{i+1,j},L_{i+1,j+N/2^{i}}+R_{i,j+N/2^{i}}\right)\\ L_{i,j+N/2^{i}}=f\left(L_{i+1,j},R_{i,j}\right)+L_{i+1,j+N/2^{i}}\\ R_{i+1,j}=f\left(R_{i,j},L_{i+1,j+N/2^{i}}+R_{i,j+N/2^{i}}\right)\\ R_{i+1,j+N/2^{i}}=f\left(L_{i+1,j},R_{i,j}\right)+R_{i,j+N/2^{i}} \end{array},\right.$$
where the function $f\left(x,y\right)=\ln\left(1+e^{x+y}\right)/\left(e^{x}+e^{y}\right)$. The $L_{n+1,j}$ messages at the rightmost stage are obtained by the channel output $\mathrm{LLR}(y_{j})$. Additionally, the $R_{1,j}$ messages at the leftmost stage are initialized by a priori known information from the information bits and frozen bits, as shown in Equations (Equation 3) and (Equation 4).
(3)$$L_{n+1,j}=\mathrm{LLR}\left(y_{j}\right),$$
(4)$$R_{1,j}=\left\{\begin{array}{cc} 0, & \mathrm{if}\;j\in I\\ +\infty , & \mathrm{if}\;j\in I^{C} \end{array}.\right.$$

Conventional flooding scheduling is applied [23]. The messages, starting from the rightmost stage, are iteratively updated column by column from right to left and then from left to right until a maximum iteration number $I_{max}$ is reached or a predefined stopping condition is valid. The BP decoder outputs the estimated vector ${\hat{u}}_{1}^{N}=\left({\hat{u}}_{1},{\hat{u}}_{2},...,{\hat{u}}_{N}\right)$ through ${\hat{u}}_{j}=\mathrm{sgn}\left(L_{1,j}+R_{1,j}\right)$, where sgn(x)=0 when x≥0, and sgn(x)=1 otherwise.

### 2.3. Belief Propagation Correction Decoder

The BPC decoder, proposed in [22], considers the perturbation on code bits located at the rightmost stage of the factor graph, which are corrupted by channel noise. When a decoding failure occurs in the original BP decoder, the decoding attempts after correction are performed to generate a valid estimate. Specifically, the correction operations assign both +∞ and −∞ to the $L_{n+1,j}$ of identified code bits in turn, and additional decoding attempts are executed until the estimate fulfills the error detection and termination check (EDTC) criterion. Each decoding attempt involves a one-bit correction. Compared with the flipping on information bit, the simulation results show that the correction on code bit has a greater impact on the decoding result. The correction set S is established based on the reliabilities of the left-propagating message determined by the channel LLRs. The decoder selects *T* indices with the lowest magnitude of $L_{n+1,j}$, i.e., $S\leftarrow j\in \{1,2,...,N\}\;\mathrm{of}\;T\;\mathrm{smallest}\;|L_{n+1,j}|$. The stopping tree [22] is utilized to reduce the sorting range. The CRC is selected as the EDTC. Let the symbol in bold denote the matrix for LLR messages. Algorithm 1 gives a brief description of the BPC decoding.
**Algorithm 1** The BPC decoding algorithm.**Input:** 
$\mathrm{LLR}(y_{1}^{N})$, $I_{max}$, *T*, I.
 1:Initialize **L** and **R** by Equations (Equation 3) and (Equation 4) 2:${\hat{u}}_{1}^{N}$← conventional BP decoding with CRC termination and $I_{max}$. 3:**if** CRC(${\hat{u}}_{1}^{N}$) = true **then** 5:   **return** ${\hat{u}}_{1}^{N}$. 5:**else** 6:   Store the messages of current state in vectors $L^{\prime }$ and $R^{\prime }$. 7:   Obtain the correction set $S=\{j_{1},j_{2},...,j_{T}\}$. 8:   **for** k=1 to *T* **do** 9:    Refresh the L and R by $L^{\prime }$ and $R^{\prime }$. And assign the ±∞ to $L_{n+1,j_{k}}$ in turn.10:    Perform additional BP iterations with CRC termination and get ${\hat{u}}_{1}^{N}$.11:    **if** CRC(${\hat{u}}_{1}^{N}$) = true **then**12:   ****return** ${\hat{u}}_{1}^{N}$.**13:    **end if**14:   **end for**15:**end if**
**Output:** 
${\hat{u}}_{1}^{N}$.


## 3. Design of Higher-Order Belief Propagation Correction Decoder

As the SNR increases, the decoding failure is gradually dominated by the graph structure, and the scheme for selecting code bits based on the value of LLR alone is flawed. This section first proposes a metric to effectively select the corrected code bit. Differently from the reduction of the searching range in [22], the stopping tree is selected as a part of the metric. Then, to solve the issue of additional latency caused by bilateral attempts, the original BPC is modified by refining the correction operation with a threshold. Then, a high-order modified BPC decoder is designed to further improve the BLER performance.

### 3.1. A Metric to Establish the Correction Set S

Inspired by the fact that the channel noise and graph structure dominate the errors in BP decoding, we propose a new metric, which combines the reliability of the code bit and the characteristic of the stopping tree where the code node is located. Specifically, eruption by channel noise is the leading cause of decoding errors in low to medium SNR regions, while the graph structure dominates the decoding result in high SNR regions. Thus, the metric combines the two factors to identify the valid corrected node.

A stopping set, defined as a subset of all variable nodes in the Tanner graph realization, is one of the critical factors affecting the BP decoding result and causing error floor [24]. In the stopping set, the adjacent check nodes of each variable node are connected to the set at least twice. For polar codes, a stopping set can be shaped like a tree rooted at a single information bit with leaves at code bits and includes at least one variable node from each middle column of the graph. A stopping tree rooted with it can be easily found for each information bit. An example of such a stopping tree is shown in Figure 2 with black nodes. Conversely, for each code bit $x_{j}$, we can find the stopping trees with that specific code bit as a leaf node. Let ST(j) denote the number of stopping trees that the code bit $x_{j}$ belongs to. According to [25], the chance of recovery for the code bit with small ST(j) is higher than others by iterations. In other words, errors in the code bits with large ST(j) have more impact on the decoding results. Therefore, we select the corrected code bit according to ST(j) combined with the reliability L(j) of code bit.

Denoting the reliability of *j*-th code bit as $L\left(j\right)=R_{n+1,j}+L_{n+1,j}$, a new metric M(j) for code bit $x_{j}$ is defined in Equation (Equation 5), where α is used to select the unreliable code bit and β is a penalty indicated by a stopping tree.
(5)M(j)=α·|L(j)|+β·(n+1)·(1/ST(j)).

By sorting the resulting metrics in ascending order, i.e., $M\left(j_{1}\right)\leq M\left(j_{2}\right)\leq ...\leq M\left(j_{N}\right)$, the nodes with the smallest metrics are selected to establish the set S, i.e., $S=\{j_{1},j_{2},...,j_{T}\},|S|=T$. The values of α and β, related to the dimension and channel noise, can be optimized by off-line Monte Carlo simulation.

### 3.2. The Modified BPC Decoder

In the BPC decoder, the correction operation assigns ±∞ to the selected code bits separately and performs decoding attempts, bringing numerous latencies overhead. Since the code bits contain prior information, the correction operation is reconsidered by the parameter V, a threshold for the magnitude of L(j), which is set to identify the current state of $x_{j}$. When |L(j)|≥V, the code bit is considered reliable so that both positive and negative assignments were executed. Otherwise, the node is corrected with its opposite side based on the sign of L(j).

The modified BPC decoder based on the proposed metric and refined correction, abbreviated as MBPC, is detailed in Algorithm 2. The function sign(x) marks the sign of *x*, $\mathrm{sign}\left(x\right)=\left\{\begin{array}{cc} 1 & x\geq 0\\ -1 & x<0 \end{array}\right.$. As shown in Algorithm 2, the enhanced correction operation in lines 9–14 avoids unnecessary attempts to reduce the complexity and latency overhead. Additionally, in lines 10 and 14, a predefined finite value τ is set, rather than *∞*, which can mitigate the impact caused by a false assignment. Here, the value of τ is set to 8. Note that, in Algorithm 1, the store operation in line 6 and refresh operation in line 9 ensure that the correction is post-processed on the conventional BP and avoid error propagation caused by previous false attempts. Based on the empirical fact that it converges quickly if the input is valid in the BP decoder, we modify the refresh operation, where the L and R messages are reset to the initial state before each attempt, as shown in Algorithm 2 line 8, resulting in avoiding complexity and memory consumption. Moreover, the modified refresh operation enables the post-processing scheme to operate in *T* independent factor graphs in parallel.
**Algorithm 2** Modified BPC decoding algorithm.**Input:**$\mathrm{LLR}(y_{1}^{N})$, $I_{max}$, I, *T*.
 1:Initial L and R by Equations (Equation 3) and (Equation 4). 2:${\hat{u}}_{1}^{N}$← conventional BP decoding with CRC termination and $I_{max}$. 3:**if** CRC(${\hat{u}}_{1}^{N}$) = true **then** 4:  **return** ${\hat{u}}_{1}^{N}$. 5:**else** 6:  Construct the set $S=\{j_{1},j_{2},...,j_{T}\}$ based on M(j) in Equation (Equation 5). 7:  **for** k=1 to *T* **do** 8:   Initial L and R by Equations (Equation 3) and (Equation 4). 9:   **if** $|L(j_{k})|<V$ **then**10:   $L_{n+1,j_{k}}=-\mathrm{sign}\left(L_{n+1,j_{k}}\right)\times \tau$11:   ${\hat{u}}_{1}^{N}$← conventional BP decoding with CRC termination.12:   **else**13:   **for** a=0:1 **do**14:    $L_{n+1,j_{k}}=\left(1-2a\right)\times \tau$.15:    ${\hat{u}}_{1}^{N}$← conventional BP decoding with CRC termination.16:   **end for**17:   **end if**18:   **if** CRC(${\hat{u}}_{1}^{N}$) = true **then**19:   **return** ${\hat{u}}_{1}^{N}$.20:   **end if**21:  **end for**22:**end if****Output:**  ${\hat{u}}_{1}^{N}$


### 3.3. Higher-Order Belief Propagation Correction Decoder

In this section, we generalize the MBPC decoder to a higher-order version, called MBPC-Ω. When the conventional BP fails to pass the CRC, the decoder executes MBPC-Ω in increasing order consecutively. Similar to the SCF-Ω decoder defined in [26], the proposed MBPC-Ω decoding is performed on nested correction sets, each of which contains $T_{k}$ elements, denoted as $S_{k}$, 1≤k≤Ω. Each element consists of *k* code bits. The nested rule is designed to construct the set. Concretely, the set $S_{k}$ is established based on the preorder set $S_{k-1}$ and its corresponding decoding results. In other words, each element in $S_{k}$ contains the element in $S_{k-1}$ and an additional code bit, which was selected based on the metric of the decoding result in the decoding attempt by $S_{k-1}$.

Denoting the *r*-th element in $S_{k-1}$ by $S_{k-1,r}$, we redefine $M{\left(j\right)}^{k,r}$ as the metric of each code bit in the decoding result of a decoding attempt with $S_{k-1,r}$.
(6)$$M{\left(j\right)}^{k,r}=\alpha \cdot \left|L{\left(j\right)}^{k-1,r}\right|+\beta \cdot \left(n+1\right)\cdot \left(1/ST\left(j\right)\right),$$
where $L{\left(j\right)}^{k-1,r}$ represents the reliability of code bits corresponding to the decoding result by $S_{k-1,r}$ and $L{\left(j\right)}^{0,r}$ represents the initial message $\mathrm{LLR}(y_{j})$. For limited computational complexity, we extract $T_{k,k-1}$ elements from $S_{k-1}$, each of which selects $T_{k,k}$ code bits with the smallest metric based on Equation (Equation 6) as the *k*-th identified bit. Thus, the size of set $S_{k}$ is $T_{k}=T_{k,k-1}\times T_{k,k}$, and the maximum number of decoding attempts of order-*k* is $2^{k}\times T_{k}$. The rule to generate the correction set can be summarized as $generate\_S_{k}$, described in Algorithm 3.

The MBPC-Ω decoder is presented in Algorithm 4. For the convenience of description, we describe the enhanced correction operation in the procedure as $correction\_operation(S_{k,r})$ in Algortihm 5. For each attempt, *k* identified code bits are corrected simultaneously. The “flag” indicates the avoidable decoding attempt. The maximum number of decoding attempts is $T=\sum_{k=1}^{\Omega }2^{k}\times T_{k}$. Note that, in the case of $|L(j_{l})|\geq V$, if the decoding attempts of both sides fail the CRC, the element to which $j_{l}$ belongs is repeatedly recorded in the next order of the correction set. If none of the decoding attempts meet the requirements of the CRC until the order exceeds Ω, the decoder declares a failure.
**Algorithm 3** The procedure of $generate\_S_{k}$.1:**for** r=1 to $T_{k,k-1}$ **do**2:   $S_{tmp}\leftarrow$$T_{k,k}$ bits with smallest $M{\left(j\right)}^{k,r}$.                         // generate a size-$T_{k,k}$ set of *k*-th corrected bit3:   **for** j=1 to $T_{k,k}$ **do**4:     $S_{k,\left(r-1\right)\cdot T_{k,k-1}+j}\leftarrow S_{k-1,r}\cup S_{tmp},j$.5:   **end for**6:**end for**

**Algorithm 4** The proposed MBPC-Ω decoding algorithm.**Input:**$\mathrm{LLR}(y_{1}^{N})$, $I_{max}$, I, *T*, Ω.
 1:Initial L and R by Equations (Equation 3) and (Equation 4). 2:Perform the original BP decoder with CRC termination. 3:**if** ${\hat{u}}_{1}^{N}$ passes the CRC **then** 4: **return**
${\hat{u}}_{1}^{N}$. 5:**else** 6: Generate the set $S_{1}$ of size $T_{1}$. 7: **for** k=1 to Ω **do** 8:  **for** r=1 to $T_{k}$ **do** 9:     // each code bit in *r*-th element expressed by $j_{l}$: $\left(j_{1},j_{2},...j_{k}\right)$.10:    a=0, flag=0.11:    **while** $a<2^{k}$ **do**12:      Initialize L and R by Equations (Equation 3) and (Equation 4).13:      Perform the $correction\_operation(S_{k,r},a,\mathrm{flag})$;   *a*++.14:      **if** flag=1 **then**15:       **continue**.16:      **end if**17:      ${\hat{u}}_{1}^{N}$← conventional BP decoding with CRC termination.18:      **if** CRC(${\hat{u}}_{1}^{N}$) = true **then**19:         **return** ${\hat{u}}_{1}^{N}$.20:       **else**21:      Construct the set $S_{k+1}$ by $generate\_S_{k}$.22:       **end if**23:      **end while**24:    **end for**25:  **end for**26: **end if****Output:**${\hat{u}}_{1}^{N}$

**Algorithm 5** The procedure of $correction\_operation(S_{k,r},a,\mathrm{flag})$.1:Binary extension for $a=(b_{1},b_{2},...,b_{k})$.2:**for** l=1 to *k* **do**3:   **if** $|L(j_{l})|<V$ and $b_{l}=1$ **then**4:  flag=1; **break**.5:   **else**6:  $L_{n+1,j_{l}}=\left(2b_{l}-1\right)\times \mathrm{sign}\left(L_{n+1,j_{l}}\right)\times \tau$;  flag=0.7:   **end if**8:**end for****Output:** flag.


## 4. Numerical Results

In this section, the simulation results are verified on a binary-input additive white Gaussian noise (BI-AWGN) channel in terms of the BLER performance and decoding latency for polar codes based on the proposed schemes. Information set I is selected based on the Monte Carlo approach [1] at 2.0 dB.

First, the BLER performance of the proposed MBPC-Ω scheme is compared with that of the BPC decoder in [22] and the EBPF decoder in [16]. MBPC-Ω with order Ω=1 is a one-bit correction scheme, the same as MBPC. The parameters of the proposed metric are set to α=1.0,β=0.75 for code length N=512 and rate R=1/2. To ensure a fair comparison, the sizes of the correction set are set to T=20 in the MBPC-1 and the BPC, while the size of the flipping set in the EBPF is set to T=40. The parameters of correction sets in the MBPC-2 and the MBPC-3 decoders are set to $T_{1}=20$, $T_{2,1}=T_{3,2}=20$ and $T_{2,2}=T_{3,3}=20$, respectively. As seen in Figure 3, the MBPC-1 decoder has a better BLER performance than the EBPF decoder and achieves a slight performance gain than that of the BPC decoder in high SNR regions. The proposed higher-order algorithm can evidently effectively improve the BLER performance, e.g., at $\mathrm{BLER}=10^{-3}$, the MBPC-2 decoder performs 0.9 dB better than the original BP 0.45 dB and better than the MBPC-1. As Ω increases, the maximum number of decoding attempts increases dramatically, bringing challenges to the complexity, but the performance gain gradually decreases. Accordingly, in the further comparisons with the existing multiple flipping scheme, only MBPC-1 and MBPC-2 are studied.

The BLER performance and average number of iterations of the proposed schemes for N=2048,R=1/2 are simulated for a comparison with the original BP, BPC, higher-order GBPF proposed in [16], and multiple BPF decoder designed based on the critical set (CS) in [15] as shown in Figure 4 and Figure 5. The 24-bit CRC is used. The parameters in Equation (Equation 6) are set to α=0.65 and β=3.5. The size of the correction set in the MBPC-1 is T=20. In the higher-order scenario, the number of maximum flipping attempts in the GBPF-2 decoder and the maximum correction operations in the MBPC-2 decoder are set to T=80+20×40 and T=40+4×10×20, respectively, to keep the maximum decoding attempts the same as those for the BPF-2, i.e., T=4×|CS|,|CS|=210. As seen in Figure 4, the performance of the proposed MBPC-2 decoder is significantly improved compared with that of the conventional BP decoder. When compared with the BPF-2 and GBPF-2 decoders, the MBPC-2 decoder achieves a similar performance in low to medium SNR regions but has a 0.1 dB performance gain when $\mathrm{BLER}=10^{-5}$. Compared with the CA-SCL decoder with L=4, the gain is around 0.2 dB in almost all simulated SNR regions. In addition, the BLER of the MBPC-2 decoder can compete with the CA-SCL decoder with L=8 when SNR≤2.2 dB, but the performance degrades at high SNR regions.

The BP-based decoding complexity and latency can be reflected as the number of iterations. Figure 5 presents a comparison of the average number of iterations. As expected, the number of average iterations of the MBPC-1 decoder is approximately half that of the BPC and EBPF decoders, while the MBPC-2 decoder also reduces the complexity and latency by half compared with the BPF-2 and GBPF-2 decoders at a similar BLER performance. Specifically, the proposed MBPC-2 decoder saves around 45% of the average number of iterations at 2.0 dB than the BPF-2 decoder; also observed is that, with the increase in SNR, the average number of iterations of the proposed MBPC decoder can approach that of the conventional BP decoder.

Furthermore, the average decoding latency in the MBPC decoder defined by the clock cycles can be expressed by Equation (Equation 7) and compared with that of the CA-SCL decoder.
(7)$$L_{ave}=I_{ave}\times 2\log N,$$
where the $I_{ave}$ represents the average number of iterations. Without considering the consumption of CRC, the decoding clock cycles of the CA-SCL decoder can be represented by the conventional SCL decoder in [27] as $L_{SCL}=2N+K-2$. Observed from Table 1, the MBPC-1 decoder has a low latency overhead at all simulated points, while the MBPC-2 decoder shows a lower decoding latency than that of the CA-SCL decoder in medium to high SNR regions.

## 5. Conclusions

In this article, a new metric, combined reliability with stopping tree, was defined to effectively identify corrected code bits in the BPC decoder. A rule for the correction operation defined by a threshold is designed to reduce the number of additional decoding attempts, especially in low to medium SNR regions. A higher-order version is designed based on the nested correction set to better decrease the gap with the performance of the CA-SCL decoder. The simulation results illustrate that the proposed higher-order MBPC decoding algorithm can effectively improve BLER performance with lower average latency in medium to high SNR regions and ensure better applicability than existing BPF decoders. Compared with the CA-SCL decoder, the proposed MBPC-2 decoder achieves around a 0.2 dB performance gain with L=4 in all simulated SNR regions and competes with the CA-SCL with L=8 when SNR ≤2.2 dB. Further research directions involve latency reduction in low SNR regions and BLER performance improvement in high SNR regions. For the latency overhead caused by massive increases in the iterations at low SNR regions, the modified refresh operation enables the decoder to be implemented in a parallel version to reduce the latency for further research, in which $T_{1}$ decoders with the same structure can be executed independently with different initializations. In addition, the method for identifying the corrected locations by deep learning is also an attractive direction of study.

## Figures and Tables

**Figure 1 entropy-24-00534-f001:**
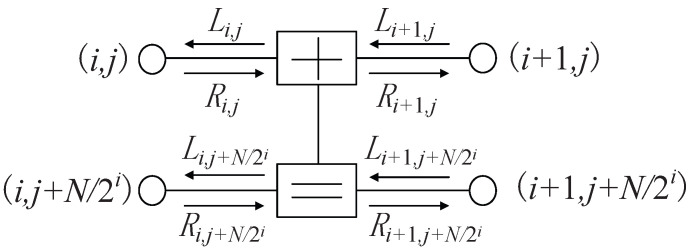
The processing element (PE) in BP decoder.

**Figure 2 entropy-24-00534-f002:**
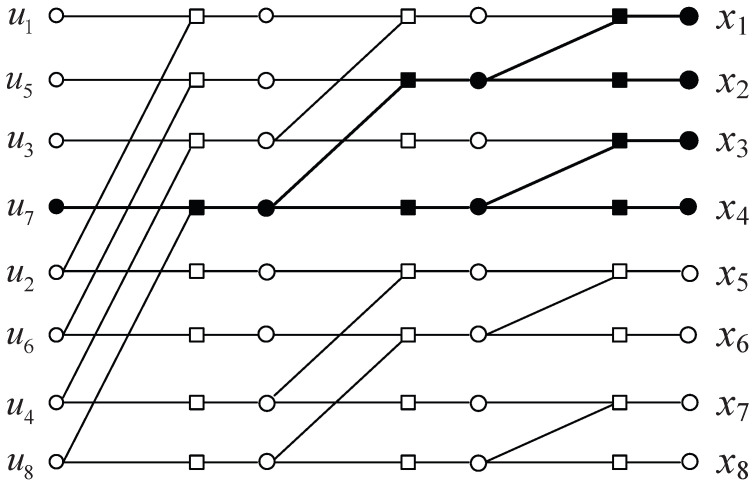
Normal realization of the encoding graph for N=8. An example of the stopping tree rooted at $u_{7}$ is shown in the black nodes.

**Figure 3 entropy-24-00534-f003:**
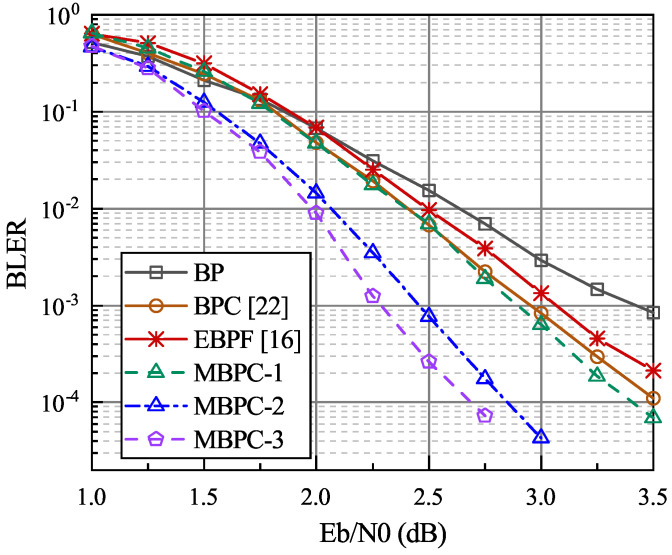
The BLER performance comparisons in polar code with length N=512, R=1/2, and $I_{max}=60$. A 16-bit CRC is used.

**Figure 4 entropy-24-00534-f004:**
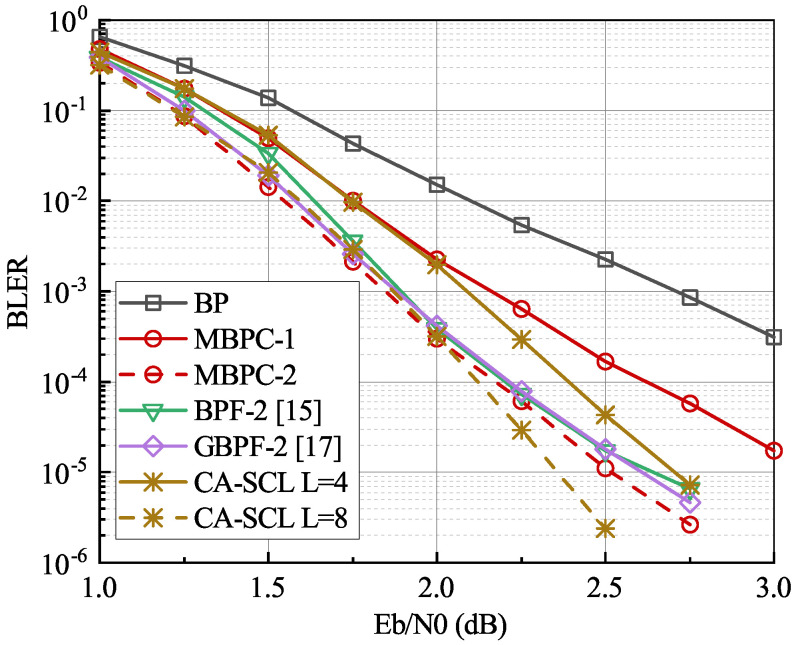
The BLER performance comparisons in the polar code with length N=2048, R=1/2, and $I_{max}=200$. A 24-bit CRC is used.

**Figure 5 entropy-24-00534-f005:**
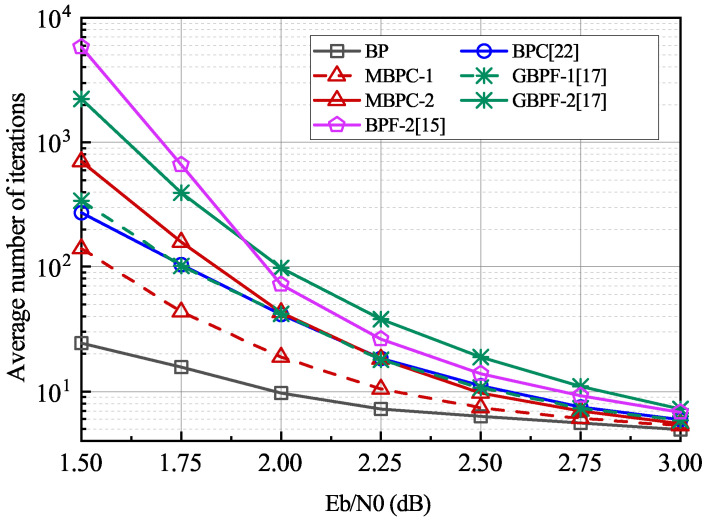
The average iteration number comparisons between the proposed MBPF and existing BP-based decoders. The code length N=2048, R=1/2 and $I_{max}=200$.

**Table 1 entropy-24-00534-t001:** Average decoding clock cycle comparisons between the proposed MBPC decoder and the CA-SCL decoder.

**Decoder**	**SNR (dB)**						
	1.5	1.75	2.0	2.25	2.5	2.75	3.0
MBPC-1	3080	946	396	220	165	133	115
MBPC-2	15,466	3476	924	396	212	154	121
	**List size**						
CA-SCL	L=4				L=8		
	5142				5142		

## Data Availability

Not applicable.

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
