# Peer review of "Higher-Order Belief Propagation Correction Decoder for Polar Codes"

_entropy, 2022, doi:10.3390/e24040534_

Round 1

Reviewer 1 Report

Channel noise and graph structure dominate the errors in BP belief propagation decoding. The authors propose a new metric combining the reliability of the code bit and the characteristic of the stopping tree where the code node is located. Extended a higher-order scheme MBPC decoding was studied for block error rate and average iteration number in comparison with the comparison with the existing decoders.

Author Response

Thank you for your comments concerning our manuscript.  The spelling and description in the manuscript were carefully checked and improved.

Reviewer 2 Report

The manuscript is well written. 

Author Response

Thank you for your comments concerning our manuscript.  The spelling and description of the methods in the manuscript were carefully checked and improved, and the revisions do not affect the schemes, results, and conclusions. 

Reviewer 3 Report

Paper entropy-1614170 “Higher-Order Belief Propagation Correction Decoder for Polar Codes”

Comments

This study proposes higher-order belief propagation correction decoder for polar codes. I think the paper fits well the scope of the journal and addresses an important subject. However, a number of revisions are required before the paper can be considered for publication. There are some weak points that have to be strengthened. Below please find more specific comments:

*Abstract: The abstract seems kind of short to me. Please provide a few sentences to clearly highlight the contributions of this work and main outcomes from the experiments.

*The authors discuss the relevant literature in the introduction section. However, there are only 16 studies acknowledged in the introduction section. Please check for more recent and relevant works. Otherwise, the literature review may not seem convincing to some readers.

*This study focuses on the utilization of propagation algorithms. I recommend for the authors to create a general discussion regarding the importance of advanced AI algorithms (e.g., heuristics, metaheuristics) for challenging decision problems. There are many different domains where advanced AI algorithms have been applied as solution approaches, such as online learning, scheduling, multi-objective optimization, transportation, medicine, data classification, and others (not just the area addressed in this study). The authors should create a discussion that highlights the effectiveness of advanced AI algorithms in the aforementioned domains. This discussion should be supported by the relevant references, including the following:

An online-learning-based evolutionary many-objective algorithm. Information Sciences 2020, 509, pp.1-21.

Two hybrid meta-heuristic algorithms for a dual-channel closed-loop supply chain network design problem in the tire industry under uncertainty. Advanced Engineering Informatics 2021, 50, p.101418.

A many-objective evolutionary algorithm with angle-based selection and shift-based density estimation. Information Sciences 2020, 509, pp.400-419.

An Optimization Model and Solution Algorithms for the Vehicle Routing Problem with a “Factory-in-a-Box”. IEEE Access 2020, 8, pp.134743-134763.

Truck scheduling optimization at a cold-chain cross-docking terminal with product perishability considerations. Computers & Industrial Engineering 2021, 156, p.107240.

Ambulance routing in disaster response considering variable patient condition: NSGA-II and MOPSO algorithms. Journal of Industrial & Management Optimization 2022, 18(2), p.1035.

Such a discussion will help improving the quality of the manuscript significantly. It can be placed somewhere within the introduction section.

*Section 2 seems to be adequate. No comments.

*The presentation of the pseudocodes in section 3 seems to be adequate.

*Description of the computational experiments could be improved. In particular, I recommend providing more references to support the selection of input data for the experiments.

*The manuscript contains several figures and tables in the computational experiments section. Please double check to make sure that all figures and tables are adequately described in the manuscript to prevent any confusion of future readers.

*The conclusions section should expand on limitations of this study and future research needs. I suggest listing the bullet points.

Author Response

Thank you for your comments concerning our manuscript.

Author Response:

Response to Reviewer 3 Comments

Point 1: Abstract: The abstract seems kind of short to me. Please provide a few sentences to clearly highlight the contributions of this work and main outcomes from the experiments.

Response 1: In this article, we proposed an modified BPC decoder, and compared it with the original BPC, the existing flipping decoders and the CA-SCL decoder. Thus, we have revised the abstract and highlighted the contributions of the proposed method. Specifically, the comparison results to the above three algorithms based on the simulation results are breifly described in the revised manuscript, respectively. Please check the revised manuscript.

Abstract:

Belief propagation (BP) decoding for polar codes has been extensively studied because of its inherent parallelism. However, its performance remains inferior to that of successive cancellation list decoding (SCL) due to the structure of the decoding graph. To improve the block error rate (BLER) performance, the BP correction (BPC) decoding, a post-processing scheme that corrects prior knowledge of the identified code bit, to improve convergence by executing additional iterations on the failed BP decoder. Moreover, the BPC decoder demonstrates a better decoding performance than the BP-based flipping decoder. Nevertheless, the additional decoding attempts lead to increased latency. In this article, a modified BPC decoder is proposed to reduce the number of decoding attempts by redefining the correction rules. A new metric is designed to effectively identify the corrected location. Numerical results show that the proposed modified BPC decoder achieves a slight improvement in BLER compared with the original BPC, with a dramatic reduction in average complexity. Furthermore, a higher-order version, named MBPC-Ω, is extended to further improve the performance, where the Ω is the maximum correction order. Numerical results show that the higher-order modified BPC achieves a similar BLER performance to existing multiple flipping BP decoders but has around half the latency overhead. In addition, the MBPC-2 decoder performs better than the cyclic redundancy check-aided SCL (CA-SCL) decoder with list size 4 and is slightly worse than the CA-SCL with list size 8 in high signal-noise ratio (SNR) regions but with significant decoding latency reduction.

Point 2: The authors discuss the relevant literature in the introduction section. However, there are only 16 studies acknowledged in the introduction section. Please check for more recent and relevant works. Otherwise, the literature review may not seem convincing to some readers.

Response 2: Based on the review’s comment, we have added some references in the revised manuscript, especially the related studies for enhanced BP decoding in the latest two years, to clearly describe the development and background of BP polar decoding, especially the improvement of the related post-processing algorithms.

Point 3: This study focuses on the utilization of propagation algorithms. I recommend for the authors to create a general discussion regarding the importance of advanced AI algorithms (e.g., heuristics, metaheuristics) for challenging decision problems. There are many different domains where advanced AI algorithms have been applied as solution approaches, such as online learning, scheduling, multi-objective optimization, transportation, medicine, data classification, and others (not just the area addressed in this study). The authors should create a discussion that highlights the effectiveness of advanced AI algorithms in the aforementioned domains. This discussion should be supported by the relevant references, including the following:

An online-learning-based evolutionary many-objective algorithm. Information Sciences 2020, 509, pp.1-21.

Two hybrid meta-heuristic algorithms for a dual-channel closed-loop supply chain network design problem in the tire industry under uncertainty. Advanced Engineering Informatics 2021, 50, p.101418.

……

Such a discussion will help improving the quality of the manuscript significantly. It can be placed somewhere within the introduction section

Response 3: The belief propagation algorithm is one of the mainstream algorithms for polar code decoding. Similar to the existing BP decoding algorithm for LDPC codes, the BP decoder for polar codes is an iterative updating decoding of soft information based on the Tanner graph realization. Different from the propagation algorithm in AI, the BP decoder for polar code has a fixed graph structure, and its initialization and information are shown in formula (2)(3) in the manuscript. The decision problems, online learning, etc. in AI are not discussed in the improved BP decoding for polar codes. Although with the development of AI, some researchers try to use the neural network in BP polar decoding, this is not what this article studies. Therefore, the introduction section does not discuss these in detail.

Deep learning was studied in BP flipping decoder to identify the error-prone bits. We have referred to it through references [19] in the revised introduction section, and the combination of AI and the proposed correction strategy has been prospected in the conclusion section.

Point 4: Description of the computational experiments could be improved. In particular, I recommend providing more references to support the selection of input data for the experiments.

Response 4: In the simulations of this article, the input data selection mainly includes the code length, code rate, the maximum size of iteration Imax, and the size of correction set T. The code length N and code rate R, which determine the code structure, aren’t the main factors in the decoding algorithm. Thus, R=1/2 in different lengths N are empirically selected for simulations. For fair comparations, the Imax and T are set to the same as the compared flipping decoding algoritihm, which is described in detail in the “Numerical Results” section.

Point 5: The manuscript contains several figures and tables in the computational experiments section. Please double check to make sure that all figures and tables are adequately described in the manuscript to prevent any confusion of future readers.

Response 5: According to the comment, we carefully check the descriptions of the figures and tables,  and ensure that all parameters in the figures and tables are declared in the manuscript.

Point 6: The conclusions section should expand on limitations of this study and future research needs. I suggest listing the bullet points.

Response 6: We have revised the conclusion section to elaborate on the improvements based on the simulation results. The further research of the proposed methods is also described in the conclusion section. Please see the revised version.

Conclusions:

In this article, a new metric, combined node reliability and stopping tree, was defined to effectively identify corrected code bits in the BPC decoder. A rule for the correction operation defined by a threshold is designed to reduce the number of additional decoding attempts, especially in low to medium SNR regions. A higher-order version is designed based on the nested correction set to better decrease the gap with the performance of the CA-SCL decoder. The simulation results illustrate that the proposed higher-order MBPC decoding algorithm can effectively improve BLER performance with lower average latency in medium to high SNR regions and ensure better applicability than existing BPF decoders. Compared with the CA-SCL decoder, the proposed MBPC-2 decoder achieves around 0.2dB BLER gains than the CA-SCL with L = 4 in all simulated SNR regions and competes with the CA-SCL with L = 8 when SNR ≤ 2.2dB. Further research directions involve latency reduction in low SNR regions and BLER performance improvement in high SNR regions. For the latency overhead caused by massive increases in the iterations in low SNR regions, the modified refresh operation enables the decoder to be implemented in a parallel version to reduce the latency for further research, in which T1 decoders with the same structure can be executed independently with different initializations. In addition, the method for identifying the corrected locations by deep learning is also an attractive studied direction.

Round 2

Reviewer 3 Report

The authors have adequately addressed my original concerns regarding the manuscript. The quality and presentation of the manuscript have been improved. Therefore, I recommend acceptance.